# Scale-Wise Visual Autoregressive Models Beat Diffusion Models on Test-Time Scaling

## Abstract

While inference-time scaling through search has revolutionized Large Language Models, translating these gains to image generation has proven difficult. Recent attempts to apply search strategies to continuous diffusion models show limited benefits, with simple random sampling often performing best. We demonstrate that the discrete, sequential nature of visual autoregressive models enables effective search for image generation. We show that beam search substantially improves text-to-image generation, enabling a 2B parameter autoregressive model to outperform a 12B parameter diffusion model across benchmarks. Systematic ablations show that this advantage comes from the discrete token space, which allows early pruning and computational reuse, and our verifier analysis highlights trade-offs between speed and reasoning capability. These findings suggest that model architecture, not just scale, is critical for inference-time optimization in visual generation.

## 1 Introduction

Generative modeling is undergoing a fundamental shift in how progress is achieved. While the past decade has been dominated by scaling model parameters (Kaplan et al., 2020; Hoffmann et al., 2022), recent breakthroughs demonstrate that inference-time computation can be equally transformative. Systems such as OpenAI's o1 (OpenAI et al., 2024) and o3 (OpenAI, 2025), DeepSeek-R1 (DeepSeek-AI et al., 2025), and Gemini 2.5 (Google, 2025) achieve substantial gains not through larger models, but via sophisticated search and deliberation during inference. For example, small language models can now match the output quality of models $14\times$ larger when given sufficient compute at test time (Snell et al., 2024). This paradigm shift raises a natural question: can visual generation similarly benefit from inference strategies rather than merely scaling up parameters?

The answer might depend on the model architecture. Recent comprehensive studies show that applying search strategies to diffusion models yields limited improvements. Approaches including noise trajectory search, zero-order optimization, and path expansion fail to outperform a simple random sampling (Ma et al., 2025). In contrast, language models benefit consistently from tree search (Yao et al., 2024; Zhang et al., 2024), reward-model guidance (Lightman et al., 2023; Wang et al., 2024; Setlur et al., 2024), and Monte Carlo methods (Xie et al., 2024). This suggests a fundamental incompatibility between continuous latent spaces and discrete search algorithms.

Visual autoregressive models, such as VAR (Tian et al., 2024) and Infinity (Han et al., 2024), generate images token by token across multiple scales, creating a search space structurally similar to that of language models. This work focuses specifically on scale-wise autoregressive models, which use next-scale prediction with limited sequential decision points, making search simpler and more computationally tractable. In this setting, algorithms like beam search can efficiently explore alternatives, prune low-quality paths early, and reuse computation across branches. By contrast, diffusion models operate in continuous noise spaces where such discrete search is less effective.

To investigate this, we conduct the first systematic study of search algorithms applied to autoregressive image generation. We combine state-of-the-art discrete visual models with verifiers ranging from lightweight preference models (Xu et al., 2023) to powerful vision-language models (Li et al., 2024), enabling guided exploration toward high-quality, compositionally accurate images.

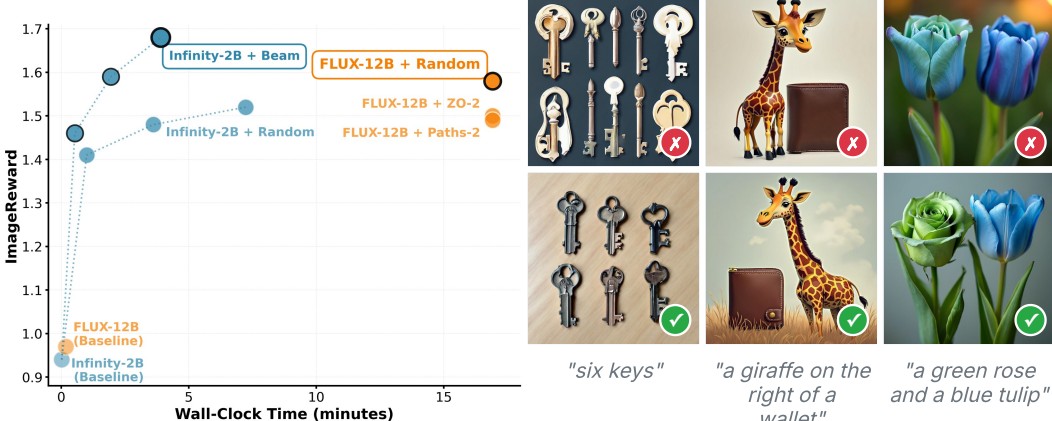

Figure 1: **Guided search in autoregressive models provides a more efficient path to high-quality image generation than scaling diffusion models.** (Left) ImageReward score vs. inference-time compute budget (wall-clock time). A 2B autoregressive model with beam search (blue) surpasses a 12B FLUX.1-dev model (Ma et al., 2025) using random search (orange), while requiring less computation. (Right) Examples showing how beam search corrects compositional errors in baseline generations. It successfully fixes object counts ("six keys"), incorrect spatial relationships ("giraffe on the right"), and color errors ("green rose and a blue tulip").

Our results challenge conventional assumptions about model scaling, showing that architectural compatibility with search can be as important as parameter count. On DrawBench and T2I-CompBench++, a 2B parameter autoregressive model with beam search achieves performance gains twice as large as those obtained by applying search to a 12B diffusion model, while using 46% fewer function evaluations. Notably, this allows the smaller model to surpass the larger model's absolute performance on all compositional metric, demonstrating that architectural compatibility with search can overcome a 6× parameter disadvantage. Fig. 1 presents both quantitative results and qualitative examples that illustrate the effectiveness of our approach.

## 2    RELATED WORK

**Visual Generation.** Diffusion models currently dominate high-quality image synthesis, achieving state-of-the-art results (Ho et al., 2020; Rombach et al., 2022; Peebles & Xie, 2023) and extending to video generation (OpenAI, 2024). Autoregressive approaches offer a different paradigm but have historically faced computational and architectural challenges. PixelRNN (Van Den Oord et al., 2016) suffered from severe computational limitations, while DALL-E (Ramesh et al., 2021) used discrete tokens but retained inefficient raster-scan ordering that disrupted spatial locality and violated bidirectional dependencies. Recent advances address these limitations through hierarchical generation strategies. Visual AutoRegressive modeling (VAR) (Tian et al., 2024) introduces "next-scale prediction" that generates images coarse-to-fine across multiple resolutions, while Infinity (Han et al., 2024) extends this with bitwise tokenization to enable infinite vocabularies. These developments establish autoregressive models as competitive alternatives to diffusion approaches while preserving their discrete, sequential nature. This property creates natural compatibility with the search algorithms that have proven effective in language modeling (Sun et al., 2024).

**Inference-Time Scaling in LLMs.** Large language models (LLMs) achieve significant performance improvements through sophisticated inference-time computation (Snell et al., 2024; OpenAI et al., 2024; DeepSeek-AI et al., 2025). Models like OpenAI's o1 (OpenAI et al., 2024) and DeepSeek-R1 (DeepSeek-AI et al., 2025) use reinforcement learning to extend internal reasoning processes, while approaches like s1 (Muennighoff et al., 2025) force computational budgets through generation control. External methods offer greater flexibility as post-training enhancements, employing tree search algorithms (Yao et al., 2024; Zhang et al., 2024) and process reward models (Lightman et al., 2023; Wang et al., 2024) to evaluate reasoning steps and guide exploration. This approach demonstrates that strategic computation can substitute for scale. Small models with compute-optimal infer-

Figure 2: **Search Strategies for Autoregressive Image Generation.** (a) Random search generates $n$ images independently and selects the one with the highest score. (b) Greedy token optimization operates sequentially; at each step, it generates $c$ candidate images and commits to the single token that produces the best result before continuing. (c) Beam search maintains $w$ parallel sequences. At each step, it explores $c$ options for each beam and keeps the top $w$ overall sequences, allowing it to explore more diverse paths than the greedy approach. All search strategies verify complete 1024×1024 images only.

ence scaling can match the performance of models up to $14\times$ larger (Snell et al., 2024), establishing that how models "think" during inference can be as important as their parameter count.

**Inference-Time Scaling in Image Generation.** Translating the success of LLM inference-time scaling to image generation has proven challenging. Recent work demonstrates that diffusion models can benefit from reward-model guidance through full model fine-tuning (Black et al., 2023; Fan et al., 2023), but this requires costly retraining of the entire model. For diffusion models operating in continuous latent spaces, recent comprehensive studies show that noise trajectory search strategies are consistently outperformed by simple random sampling (Ma et al., 2025), revealing fundamental architectural limitations. Ma et al. (2025) provide the most comprehensive and state-of-the-art evaluation of test-time search strategies for diffusion models, systematically comparing multiple search algorithms across different verifiers. While frameworks like Feynman-Kac steering (Singhal et al., 2025) and flow model adaptations (Kim et al., 2025) show some promise, gains remain limited compared to LLM achievements. Progress has largely been confined to preference optimization (Wallace et al., 2023; Tong et al., 2025), rejection sampling with VLM-based selection (Xie et al., 2025), or reflection-based approaches (Zhuo et al., 2025). Notably, recent work explores chain-of-thought reasoning for autoregressive image generation (Guo et al., 2025), and concurrent work on TTS-VAR (Chen et al., 2025) explores test-time scaling for VAR models through adaptive batch sampling and clustering-based diversity search, but the full potential of search strategies remains unexplored. This contrast between limited gains in continuous models and huge improvements in discrete LLMs suggests that architectural compatibility with search algorithms may be the critical missing ingredient for effective inference-time scaling in visual generation.

## 3 METHOD

We apply tree search algorithms to autoregressive image generation, exploiting the discrete token structure to enable efficient guided exploration. We describe the base generative model (Sec. 3.1), our verification framework (Sec. 3.2), and search strategies (Sec. 3.3).

### 3.1 AUTOREGRESSIVE IMAGE GENERATION

We employ Infinity (Han et al., 2024), a state-of-the-art autoregressive model that fundamentally differs from traditional autoregressive image generation. While models like VQGAN (Esser et al., 2021) and LlamaGen (Sun et al., 2024) generate images token-by-token in raster-scan order—requiring sequential generation of thousands of tokens, Infinity employs *scale-wise* generation.

Specifically, Infinity generates $1024 \times 1024$ images through 13 progressive scales using "next-scale prediction": $p(\mathbf{R}) = \prod_{k=1}^{K} p(r_k | r_1, \ldots, r_{k-1})$, where each $r_k$ represents *all* tokens at scale $k$, generated simultaneously in a single forward pass. This creates a fundamental advantage with only 13 decision points, and the discrete token generation enables computational reuse. Once tokens at scales $r_1, \ldots, r_k$ are computed, their transformer key-value representations can be cached and reused across all search branches sharing that prefix.

## 3.2 VERIFICATION FRAMEWORK

Effective search requires reliable quality assessment at multiple scales. We employ complementary verification strategies to capture different aspects of image generation quality. ImageReward (Xu et al., 2023) serves as our primary verifier, providing a lightweight human preference predictor.

To ensure comprehensive evaluation beyond a single metric, we also include CLIPScore (Hessel et al., 2021) and Aesthetic Score (Schuhmann et al., 2022), which assess semantic alignment and visual quality independent of prompt adherence, respectively. Since these three metrics capture fundamentally different aspects (i.e., semantic correspondence, visual appeal, and human preference), we also use an ensemble verifier that combines their assessments through ranking-based aggregation. The ensemble computes each candidate's rank according to each verifier and selects based on the average ranking. This verification follows (Ma et al., 2025) and allows for robust comparison with a similar approach using diffusion models. Additionally, we employ LLaVA-OneVision (Li et al., 2024; Guo et al., 2025), a 7B vision-language model fine-tuned for prompt alignment assessment. Unless otherwise stated, ImageReward is used as a tiebreaker if LLaVA-OneVision returns multiple candidates with the same score.

## 3.3 SEARCH STRATEGIES

We compare three search strategies with distinct exploration-exploitation trade-offs (Fig. 2):

**Random Search.** Generates $n$ complete images independently using different random seeds, selecting the highest-scoring result: $\mathbf{R}^* = \arg\max_{i \in [n]} S(\mathbf{R}^{(i)})$. This provides maximal diversity but cannot exploit the sequential generation structure for computational efficiency.

**Greedy Token Optimization (GTO).** At each scale $k$, generates $c$ complete continuations from the current prefix, selecting the token that produces the highest-scoring final image:

$$r_k^* = \arg\max_{j \in [c]} S(r_1, \ldots, r_{k-1}, r_k^{(j)}, r_{k+1}^{(j)}, \ldots, r_K^{(j)}) \tag{1}$$

Crucially, at each scale $k$, we generate complete images through all 13 scales before verification, the verifier evaluates only finished 1024×1024 images, never partial generations. This creates a single optimized path but risks local optima. We evaluate $c \in \{4, 15, 30\}$ spanning computationally efficient to more thorough exploration regimes.

**Beam Search.** Maintains $w$ parallel hypotheses, expanding each with $c$ candidates at every scale. After scoring all $w \times c$ complete images, it retains the top $w$ prefixes for continued expansion:

$$\text{Beams}_{k+1} = \text{top-}w\{S(\mathbf{R}) : \mathbf{R} \in \text{Candidates}_k\} \tag{2}$$

This balances exploration breadth with computational tractability through aggressive pruning. We evaluate $(w, c) \in \{(2, 2), (3, 5), (3, 10)\}$, ranging from minimal to extensive search configurations.

The computational advantage of GTO and beam search stems from prefix reuse: shared computation reduces complexity from $O(n \cdot K)$ for independent generation to approximately $O(n \cdot K/w)$ for beam search with width $w$, where $n$ represents candidate images and $K$ the number of scales.

## 3.4 COMPUTATIONAL BUDGET METRICS

Fair comparison requires precise measurement of computational cost, particularly given the asymmetric efficiency gains enabled by discrete search. We report three complementary metrics:

**Number of Images.** Total complete images evaluated by the verifier. This provides direct comparability with prior best-of-N studies (Guo et al., 2025; Xie et al., 2025; Ma et al., 2025) regardless of architectural differences.

**Number of Function Evaluations (NFEs).** Total transformer forward passes, where one NFE corresponds to generating tokens for one of Infinity's 13 scales. This metric reveals the true computational advantage of guided search: beam search, generating 195 images requires only 1,365 NFEs versus 2,535 NFEs for equivalent random search, reflecting the 46% efficiency gain achieved.

**Wall-Clock Time.** Generation time in seconds on a single H100 GPU. When applicable, we report verification overhead separately.

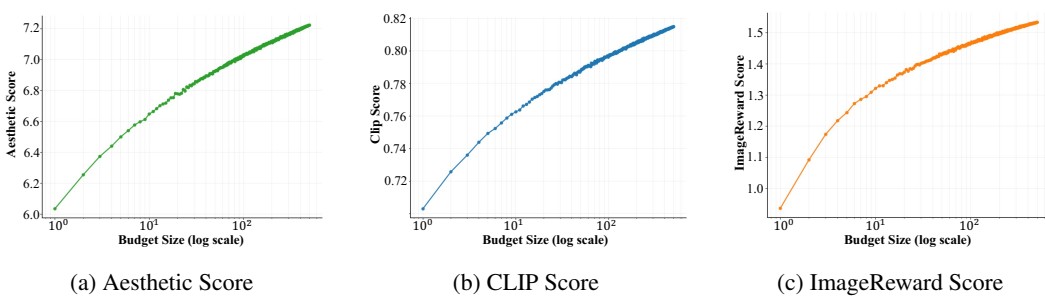

(a) Aesthetic Score        (b) CLIP Score        (c) ImageReward Score

Figure 3: **Expected maximum verification scores as functions of budget size (log scale).** All three verifiers exhibit logarithmic scaling.

We note that NFEs across architectures (autoregressive vs. diffusion) are not directly comparable in FLOPs but provide meaningful order-of-magnitude efficiency comparisons, while wall-clock time enables direct practical comparison.

## 4 EXPERIMENTAL RESULTS

This section evaluates our proposed search strategies for test-time scaling in discrete autoregressive image generation. We first analyze how verification scores scale with computational budget (Sec. 4.1), then demonstrate beam search's superior performance on DrawBench (Saharia et al., 2022) (Sec. 4.2). We examine verifier trade-offs (Sec. 4.3) and validate on compositional benchmarks T2I-CompBench++ (Huang et al., 2025), DPG-Bench (Hu et al., 2024), and GenEval Ghosh et al. (2023) (Sec. 4.4). Finally, we compare against inference-time scaling in continuous diffusion models (Ma et al., 2025), showing that discrete search overcomes a $6\times$ parameter deficit (Sec. 4.5).

**Implementation details.** Unless otherwise stated, all experiments use the Infinity-2B model (Han et al., 2024) with bitwise VQ-VAE vocabulary $V_d = 2^{32}$ and patch size 16, generating $1024 \times 1024$ images through 13 progressive scales. We set temperature $\tau = 1.0$ and cfg parameter to 3.0, with Flan-T5-XL (Chung et al., 2022) as the text encoder. When Infinity-8B is employed (explicitly noted), the configuration uses VQ-VAE vocabulary $V_d = 2^{56}$ with patch size 8 and spatial patchification enabled. All experiments run on a single Nvidia H100 GPU with bf16 precision.

### 4.1 BUDGET SCALING ANALYSIS

To understand the fundamental trade-offs in test-time compute allocation, we first investigate how verification scores scale with computational budget in a simple search setting. This analysis establishes the baseline scaling behavior that motivates more sophisticated search strategies.

**Experimental setup.** We generate 500 samples per prompt using Infinity-2B on 200 prompts from DrawBench, evaluating each sample with ImageReward (Xu et al., 2023), Aesthetic Score (Schuhmann et al., 2022), and CLIPScore (Hessel et al., 2021). For each budget size $k \in \{1, 2, ..., 500\}$, we compute the expected maximum score by randomly sampling $k$ images from the full set and selecting the highest-scoring sample, repeating this process 10 times to obtain robust statistics.

**Scaling behavior.** Fig. 3 reveals a clear logarithmic relationship between budget size and expected maximum verification score across all three verifiers: $\mathbb{E}[\max_{i \leq k} s_i] \approx \alpha \log(k) + \beta$, where $s_i$ denotes the verification score of sample $i$, and $\alpha, \beta$ are verifier-specific constants. This logarithmic scaling indicates diminishing returns: achieving each additional unit of quality improvement requires exponentially more samples.

This scaling law establishes a fundamental limitation of random search and motivates the development of more efficient strategies. While increasing the sampling budget consistently improves quality, the logarithmic relationship means that substantial quality gains require exponentially larger computational investments, motivating the use of search algorithms for inference-time scaling.

| Verifier | Imgs. | NFEs | Time (s) | CLIPScore | | | ImageReward | | | Aesthetic | | | Ensemble | | |
|---|---|---|---|---|---|---|---|---|---|---|---|---|---|---|---|
| | | | | Aes. | CLIP | ImgR. | Aes. | CLIP | ImgR. | Aes. | CLIP | ImgR. | Aes. | CLIP | ImgR. |
| Baseline (2B) | - | 13 | 1.1 | 6.06 | 0.71 | 0.94 | 6.06 | 0.71 | 0.94 | 6.06 | 0.71 | 0.94 | 6.06 | 0.71 | 0.94 |
| Baseline (8B) | - | 13 | 2.1 | 5.97 | 0.73 | 1.14 | 5.97 | 0.73 | 1.14 | 5.97 | 0.73 | 1.14 | 5.97 | 0.73 | 1.14 |
| Random | 390 | 5070 | 434 | 5.83 | 0.80 | 1.09 | 6.11 | 0.72 | 1.52 | 7.19 | 0.67 | 0.87 | 6.53 | 0.75 | 1.28 |
| Random | 195 | 2535 | 217 | 5.88 | 0.79 | 1.13 | 6.05 | 0.73 | 1.48 | 7.08 | 0.68 | 0.88 | 6.31 | 0.72 | 1.16 |
| Random | 54 | 702 | 60 | 5.91 | 0.78 | 1.08 | 6.06 | 0.72 | 1.41 | 6.92 | 0.68 | 0.88 | 6.31 | 0.73 | 1.21 |
| GTO | 390 | 2730 | 234 | 5.88 | 0.85 | 1.08 | 6.08 | 0.73 | 1.62 | 7.50 | 0.68 | 0.90 | 6.76 | 0.78 | 1.44 |
| GTO | 195 | 1365 | 117 | 5.93 | 0.83 | 1.08 | 6.08 | 0.73 | 1.58 | 7.38 | 0.69 | 0.93 | 6.66 | 0.77 | 1.37 |
| GTO | 54 | 377 | 32 | 6.01 | 0.79 | 1.07 | 6.09 | 0.72 | 1.45 | 6.98 | 0.69 | 0.91 | 6.50 | 0.75 | 1.31 |
| Beam search | 390 | 2730 | 234 | 5.86 | **0.86** | 1.16 | 6.16 | 0.73 | **1.68** | **7.75** | 0.67 | 0.95 | **6.92** | **0.79** | **1.50** |
| Beam search | 195 | 1365 | 117 | 5.84 | 0.83 | 1.10 | 6.10 | 0.73 | 1.59 | 7.38 | 0.69 | 0.96 | 6.68 | 0.78 | 1.44 |
| Beam search | 54 | 377 | 32 | 5.93 | 0.79 | 1.07 | 6.09 | 0.72 | 1.46 | 6.94 | 0.69 | 0.95 | 6.46 | 0.75 | 1.34 |

Table 1: **Comparison of search strategies on DrawBench.** We show Random search, Greedy Token Optimization (GTO), and Beam search with varying computational budgets. "Imgs." indicates verified images during search and "NFEs" the total function evaluations. Each verifier column displays results when that verifier guides the search, measuring Aesthetic (Aes.), CLIPScore (CLIP), and ImageReward (ImgR.). Advanced strategies outperform random search even with reduced budgets. Bold indicates best scores per verifier.

## 4.2 PERFORMANCE ON DRAWBENCH

We present our comparison of search strategies on the DrawBench benchmark, which comprises 200 carefully curated text prompts distributed across distinct categories. We evaluate the performance of random search, Greedy Token Optimization (GTO), and beam search when guided by different verifiers. Tab. 1 presents the results on DrawBench. While all tested approaches outperform the baseline Infinity-2B model, beam search is the most effective method, delivering the strongest improvements across different quality metrics. The optimized smaller model not only achieves gains in visual quality, prompt adherence, and aesthetic appeal, but surpasses the performance of the larger Infinity-8B baseline. For a qualitative comparison of search strategies on DrawBench, see Appendix B.

Each verifier exhibits distinct optimization characteristics. ImageReward guidance produces the largest gains while preserving performance on other metrics. CLIPScore optimization improves both CLIP and ImageReward scores but sacrifices aesthetic quality. Aesthetic Score optimization enhances visual appeal but degrades prompt adherence (measured by CLIPScore), demonstrating the verifier hacking phenomenon where search processes overfit to narrow objectives.

Advanced search strategies show superior computational efficiency. Both GTO and beam search outperform random search while using substantially fewer resources. Beam search with 195 images (1,365 NFEs) surpasses random with 390 images (5,070 NFEs), achieving this efficiency gain through prefix caching and guided exploration. This efficiency persists until extremely low budgets, where beam search with only 54 images delivers competitive results despite using just 7% of random search's cost. We further explore a heuristic dynamic budget allocation in Appendix D.

## 4.3 VERIFIER ANALYSIS

Having shown that guided search consistently outperforms random search, we now analyze verifier trade-offs to understand when each is most effective. Appendix C shows how different verifiers affect scaling behavior under varying computational budgets.

**Computational requirements.** The computational overhead of different verifiers varies dramatically, as shown in Tab. 2. Using images generated by Infinity-2B, we find substantial differences in both processing time and memory usage. While lightweight verifiers like CLIPScore process images in just 14ms using

| Verifier | Time/Img (ms) | GPU Mem (GB) |
|---|---|---|
| CLIPScore | 14 | 1.6 |
| Aesthetic | 19 | 1.6 |
| ImageReward | 25 | 1.7 |
| LLaVA-OneVision | 500 | 15.3 |

Table 2: **Verifier computational requirements.** Results show up to 36× speed difference and 9× memory difference between lightweight verifiers and LLaVA-OneVision.

~1.6GB GPU memory, LLaVA-OneVision requires 500ms per image and 15.3GB GPU memory,

| Verifier | Imgs. | NFEs | Time (s) | Color | Shape | Texture | Spatial | Numeracy | Complex |
|---|---|---|---|---|---|---|---|---|---|
| Baseline | - | 13 | 1.1 | 0.75 | 0.47 | 0.61 | 0.26 | 0.54 | 0.39 |
| ImageReward + Random | 195 | 2535 | 217 (+5) | 0.84 | 0.62 | 0.74 | 0.27 | 0.61 | 0.41 |
| ImageReward + Random | 105 | 1365 | 117 (+3) | 0.82 | 0.61 | 0.73 | 0.28 | 0.59 | 0.41 |
| ImageReward + Beam | 195 | 1365 | 117 (+5) | **0.84** | **0.63** | **0.75** | 0.30 | 0.62 | **0.42** |
| LLaVA-OneVision + Random | 195 | 2535 | 217 (+98) | **0.84** | 0.61 | 0.75 | 0.35 | 0.64 | 0.41 |
| LLaVA-OneVision + Random | 105 | 1365 | 117 (+53) | 0.83 | 0.60 | 0.75 | **0.37** | 0.64 | 0.41 |
| LLaVA-OneVision + Beam | 195 | 1365 | 117 (+98) | 0.83 | **0.64** | **0.76** | 0.36 | **0.67** | **0.42** |

Table 4: **Compositional performance comparison on T2I-CompBench++.** Task-specific performance when using different search strategies with varying computational budgets. Beam search consistently improves performance while using only 54% of random search's NFE budget and achieving close to 2× speedup in wall-clock time. Times shown as generation + verification overhead (in parentheses). Bold indicates best performance per category.

representing a 36× speed difference and 9× memory overhead. Notably, the verification time with LLaVA-OneVision approaches the generation time of Infinity-2B (800ms), making verifier choice a critical bottleneck in practical deployment.

Tab. 3 presents verifier substitution analysis, measuring actual task performance when using different verifiers for best-of-195 selection on T2I-CompBench++. The results reveal a clear trade-off between lightweight and heavyweight verifiers. For attribute-binding tasks, ImageReward consistently outperforms LLaVA-OneVision, showing advantages of 0.02 on color, 0.02 on shape, and 0.02 on texture. However, LLaVA-OneVision provides a decisive 0.07 advantage on spatial reasoning tasks, where its vision-language reasoning capabilities are essential. Given ImageReward's 20× computational advantage, these results suggest task-dependent verifier selection: lightweight models for attribute binding, heavyweight models for complex reasoning.

A prevalent issue when scaling inference through search is verifier hacking, where the search process overfits to a verifier's inherent biases. This phenomenon arises because verifiers often have narrow objectives; optimizing for one metric can degrade another. For instance, an aesthetic verifier might ignore

| Verifier | Color | Shape | Texture | Spatial | Numeracy | Complex |
|---|---|---|---|---|---|---|
| Baseline | 0.75 | 0.47 | 0.61 | 0.26 | 0.54 | 0.39 |
| Aesthetic | 0.74 | 0.47 | 0.58 | 0.23 | 0.54 | 0.39 |
| CLIP | 0.79 | 0.54 | 0.70 | 0.26 | 0.58 | 0.40 |
| ImageReward | **0.84** | **0.62** | **0.74** | 0.27 | 0.61 | **0.41** |
| Ensemble | 0.81 | 0.58 | 0.70 | 0.29 | 0.61 | 0.40 |
| LLaVA-OneVision* | 0.82 | 0.60 | 0.72 | **0.36** | **0.62** | 0.41 |

Table 3: **Verifier substitution on T2I-CompBench++.** Task-specific performance when using each verifier for best-of-195 selection, evaluated using the T2I-CompBench pipeline. Bold indicates best performance per category. * used without tie-breaking.

key prompt details to generate a more visually appealing image, while a CLIPScore-guided search might sacrifice visual quality for strict prompt adherence. For compositional tasks, we see the best results, and least amount of hacking, for ImageReward and LLaVa-OneVision (see Appendix E for a qualitative example).

## 4.4 COMPOSITIONAL VALIDATION

Having established search effectiveness on DrawBench and analyzed verifier trade-offs, we now evaluate our approach on the more demanding compositional challenges across two specialized benchmarks. We focus on the most promising verifiers identified in our analysis: ImageReward for its cost-effectiveness and LLaVA-OneVision for its reasoning capabilities.

**T2I-CompBench++.** We compare beam search against random search using both verifiers on 1,800 prompts across six compositional categories: color, shape, texture, spatial, numeracy, and complex compositions. Both strategies use 195 images, with beam search requiring only 1,365 NFEs compared to random search's 2,535 NFEs. We employ LLaVA-OneVision with ImageReward-based tie-breaking to resolve cases where multiple images receive identical binary assessments.

Tab. 4 shows that both search strategies demonstrate substantial improvements over the baseline across all compositional categories. The baseline Infinity-2B model achieves modest scores, partic-

| Verifier | Imgs. | NFEs | Time (s) | Relation | Entity | Other | Attribute | Global | Overall |
|---|---|---|---|---|---|---|---|---|---|
| Baseline | - | 13 | 1.1 | 92.13 | 88.56 | 83.20 | 86.73 | 84.19 | 82.34 |
| ImageReward + Random | 105 | 1365 | 117 (+3) | 92.98 | 89.85 | 83.60 | 86.99 | 83.59 | 83.80 |
| LLaVA + Random | 105 | 1365 | 117 (+53) | 93.87 | 91.63 | 86.80 | **88.98** | 83.89 | 85.94 |
| ImageReward + Beam | 195 | 1365 | 117 (+5) | 92.90 | 89.79 | 84.80 | 86.91 | 83.89 | 84.41 |
| LLaVA + Beam | 195 | 1365 | 117 (+98) | **94.02** | **91.77** | **89.60** | 88.33 | **84.80** | **86.62** |

Table 5: **DPG-Bench results.** At matched generation compute (1365 NFEs, ∼117s generation), stronger verifiers prove more reliable on complex prompts. LLaVA-guided beam achieves best on 5 of 6 metrics, while ImageReward random outperforms beam on 3 of 6, demonstrating that lightweight preference-based verifiers struggle to guide search on detailed prompts. Times shown as generation + verification overhead. Bold indicates best per category at 1365 NFEs.

ularly struggling with shape (0.47) and spatial reasoning (0.26) tasks. ImageReward with random search demonstrates strong performance, achieving a color binding score of 0.84, an impressive 0.09 improvement over baseline. When paired with beam search, ImageReward shows competitive results across all categories, with notable improvements in spatial reasoning (+0.03) and numeracy (+0.01) over its random search counterpart. Qualitative examples with the performance of beam search are shown in Fig. 4 (more examples can be found in Appendix A).

Despite the 20× computational overhead of LLaVA-OneVision, performance differences between verifiers depend on task complexity. Margins are modest for simple attribute binding, but reasoning-heavy tasks show larger gaps:

| Method | One Obj. | Two Obj. | Count | Colors | Position | Color Attr. | Overall |
|---|---|---|---|---|---|---|---|
| Baseline | 1.00 | 0.78 | 0.60 | 0.85 | 0.25 | 0.55 | 0.67 |
| Beam Search | 1.00 | 0.97 | 0.85 | 0.90 | 0.51 | 0.74 | 0.83 |
| *Improvement* | *0.00* | *0.19* | *0.25* | *0.05* | *0.26* | *0.19* | *0.16* |

Table 6: **Performance comparison on GenEval.** Beam search shows substantial improvements across object-focused compositional tasks, particularly in multi-object, counting, and spatial reasoning capabilities.

LLaVA-OneVision outperforms ImageReward on spatial reasoning and numeracy, where understanding object relationships and counting is critical. LLaVA-OneVision's vision-language reasoning capabilities clearly surpass the preference-based ImageReward model in these categories.

**DPG-Bench.** To evaluate performance on longer, more detailed prompts, we test on DPG-Bench (Hu et al., 2024). Unlike the more concise prompts in GenEval and T2I-CompBench++, DPG-Bench features elaborate descriptions that test whether search strategies scale to complex text inputs. Tab. 5 shows results at matched compute (1365 NFEs). The results reveal that verifier choice becomes critical as prompt complexity increases. LLaVA-guided beam search wins on 5 of 6 metrics, with the largest gain on "other" (+2.80 points). In contrast, ImageReward-guided random search actually outperforms beam on 3 of 6 metrics (relation, entity, attribute), demonstrating that lightweight preference-based verifiers struggle to reliably guide search when prompts become detailed and compositionally demanding. Overall, we still see improvements by using beam search for both ImageReward and LLaVa.

**GenEval.** To further validate our approach on object-focused compositional evaluation, we evaluate on GenEval (Ghosh et al., 2023), an object-focused framework that evaluates compositional image properties such as position, count, and color. Tab. 6 demonstrates that beam search achieves significant improvements across all GenEval categories. The baseline Infinity-2B model shows particular weaknesses in positional reasoning (0.25) and color attribution (0.55). Beam search delivers substantial gains: +19% on

| Beam search | Color | Shape | Texture | Spatial | Numeracy | Complex |
|---|---|---|---|---|---|---|
| $\tau = 1.0$ | 0.84 | 0.63 | 0.75 | **0.30** | 0.62 | **0.42** |
| $\tau = 2.0$ | **0.85** | **0.65** | **0.76** | 0.29 | **0.64** | 0.41 |

Table 7: **Performance of ImageReward-guided beam search with different sampling temperatures.** Increasing temperature boosts performance in most categories but reveals a trade-off in spatial tasks. Experiment conducted with 195 images (1365 NFEs).

two-object composition, +25% on counting tasks, +26% on position, and +19% on color attribution.

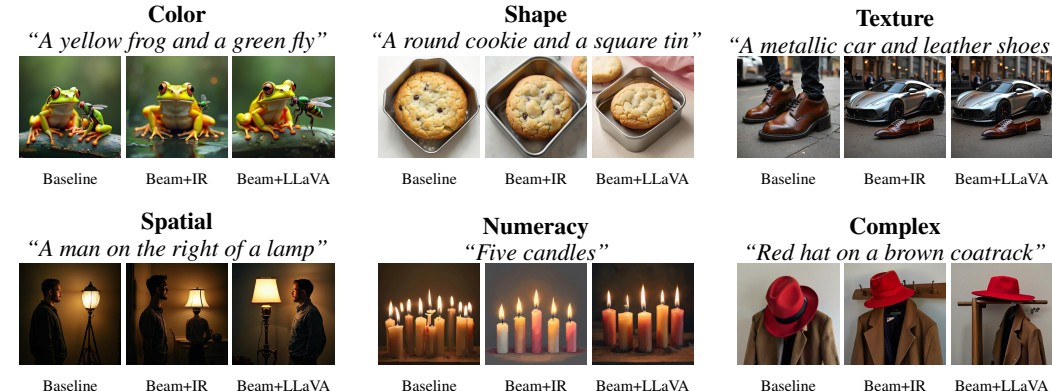

Figure 4: **Qualitative comparison of verifiers on T2I-CompBench++.** Each example shows the performance of the baseline and the beam search with ImageReward and LLaVA-OneVision. For attribute binding (e.g., Color), both verifiers perform well. For complex reasoning (e.g., Spatial), LLaVA-OneVision's capabilities are required to guide the model to a semantically correct image, correcting errors that ImageReward misses.

The overall average improvement of +16% demonstrates the effectiveness of guided search for object-focused compositional generation.

Across both benchmarks, beam search improvements remain significant, particularly considering that beam search achieves these gains using only 54% of random search's NFE budget on T2I-CompBench++. This efficiency, combined with consistent improvements across compositional metrics, establishes beam search as a practical strategy for scaling inference. The results suggest a nuanced trade-off: for attribute-focused tasks, lightweight ImageReward paired with beam search offers excellent cost-efficiency, while reasoning-heavy applications benefit from LLaVA-OneVision's superior capabilities despite computational overhead.

**The Impact of Sampling Temperature.** To explore how sampling diversity impacts search performance, we conducted an experiment increasing the generation temperature ($\tau$) from 1.0 to 2.0 for ImageReward-guided beam search. Tab. 7 shows that this change boosts performance on most tasks but reveals a nuanced trade-off rooted in the verifier's capabilities. The higher temperature provides more diverse candidate paths, particularly beneficial for numeracy tasks (+2.0%), likely because increased variety allows the model to produce different counts from which ImageReward can effectively select the correct image. Conversely, the slight decrease in spatial performance likely amplifies ImageReward's inherent weakness in this domain, as increased diversity may produce more spatially varied but incorrect layouts that the verifier struggles to penalize effectively.

### 4.5 COMPARISON WITH CONTINUOUS DIFFUSION MODELS

We now compare our results with Ma et al. (2025), who explore inference-time scaling for continuous diffusion models. Their experiments utilize FLUX.1-dev, a 12B parameter model, while our approach employs the Infinity-2B model (2B parameters), a $6\times$ difference in model capacity.

**General-purpose validation on DrawBench.** Tab. 8 provides a detailed comparison of our beam search strategy against the best-performing search from Ma et al. (2025) on DrawBench. The data highlights two critical advantages of our approach. First, our method is substantially more efficient: our medium-budget beam search (1365 NFEs) achieves a

| Method | NFEs | Imgs. | Time (s) | Aes | CLIP | ImgR. |
|---|---|---|---|---|---|---|
| Ma et al. (2025) | | | | | | |
| FLUX.1-dev-12B | - | 1 | 11 | 5.79 | 0.71 | 0.97 |
| + Random search (Best) | 2880 | 96 | 1015 | 6.38 | 0.82 | 1.58 |
| *Absolute Gain* | | | | *+0.59* | *+0.11* | *+0.61* |
| *Our Method* | | | | | | |
| Infinity-2B | - | 1 | 1.1 | 6.06 | 0.71 | 0.94 |
| + Beam search (Med) | 1365 | 195 | 117 | 7.38 | 0.83 | 1.59 |
| + Beam search (High) | 2730 | 390 | 234 | **7.75** | **0.86** | **1.68** |
| *Absolute Gain (High)* | | | | *+1.69* | *+0.15* | *+0.74* |

Table 8: **Performance and efficiency comparison on DrawBench.** Our 2B model with beam search surpasses the 12B FLUX.1-dev model across all metrics. Our medium-budget setting already exceeds the competitor's performance while using less than half the NFEs and achieving significantly faster wall-clock time. Best final scores and gains are in **bold**.

| Method | Model | NFEs | Imgs. | Time | T2I-CompBench Categories | | | | | |
|--------|-------|------|-------|------|-------|-------|---------|---------|----------|---------|
| | | | | (s) | Color | Shape | Texture | Spatial | Numeracy | Complex |
| Ma et al. baseline | FLUX.1-dev (12B) | - | 1 | 11 | 0.77 | 0.52 | 0.63 | 0.24 | 0.62 | 0.36 |
| Ma et al. + Search | FLUX.1-dev (12B) | 1920 | 64 | 677 | 0.82 | 0.60 | 0.72 | 0.30 | 0.66 | 0.38 |
| *Absolute gain* | | | | | 0.05 | 0.08 | 0.09 | 0.06 | 0.05 | 0.02 |
| Our baseline | Infinity-2B (2B) | - | 1 | 1.1 | 0.75 | 0.47 | 0.61 | 0.26 | 0.54 | 0.39 |
| Our + Beam search | Infinity-2B (2B) | 1365 | 195 | 117 | 0.83 | 0.64 | 0.76 | 0.36 | 0.67 | 0.42 |
| *Absolute gain* | | | | | **0.08** | **0.17** | **0.15** | **0.10** | **0.13** | **0.04** |

Table 9: **Comparison with Ma et al. (2025) on T2I-CompBench++.** The discrete autoregressive approach achieves superior absolute improvements despite using a 6× smaller model. This superior performance is achieved with fewer total function evaluations (1365 NFEs vs. 1920 NFEs) and 5.8× faster wall-clock time (117s vs. 677s), underscoring the computational efficiency of guided discrete search. Bold indicates better improvement metrics.

higher ImageReward score than the competitor's best result, using less than half the computational budget. Second, at comparable budgets, our high-budget search (2730 NFEs) outperforms the 12B model's best search result (2880 NFEs) across every metric, achieving performance gains that are 1.3× to 3.1× larger.

**Compositional validation on T2I-CompBench++.** Tab. 9 presents a comparison of absolute and relative improvements on T2I-CompBench++. Despite the substantial difference in model size, the autoregressive approach demonstrates superior performance gains across all evaluation categories. The 2B autoregressive model with beam search achieves higher scores than the 12B diffusion model with search in every compositional category, providing strong evidence that architectural compatibility with search can overcome a 6× deficit in model parameters. Despite starting from a lower baseline, we achieve substantially larger absolute improvements: an average of 11.3% across all categories compared to Ma et al. (2025) average of 5.7%. The scaling behavior is particularly evident in structured tasks like shape (+17.38% vs. +7.72%) and spatial reasoning (+10.45% vs. +6.14%). Notably, our spatial reasoning and counting results utilize the computationally expensive LLaVA-OneVision verifier, yet the smaller autoregressive model still outperforms the larger diffusion model, further underscoring that architectural advantages can overcome both parameter count and verifier computational overhead. This suggests that discrete token optimization is inherently better suited for test-time scaling on compositional tasks than continuous latent space optimization.

**Architectural advantages of discrete search.** These results demonstrate that discrete autoregressive models provide a fundamentally more tractable domain for guided search compared to continuous diffusion models. The discrete token space enables early pruning of unpromising paths and computational reuse through prefix caching. This architectural benefit leads to a step-change in performance scaling, with our approach achieving not only larger relative improvements but also higher absolute performance despite using a model with 6× fewer parameters. The superior scaling behavior suggests that for compositional image generation, architectural compatibility with search may be more important than raw parameter count, opening new directions for efficient model development.

## 5 CONCLUSION

This work demonstrates that autoregressive image models hold a fundamental architectural advantage for inference-time search. Their discrete token space enables efficient pruning and computational reuse, allowing a 2B model with beam search to surpass a 12B diffusion model while using fewer function evaluations. These gains are consistent across benchmarks, highlighting the robustness of this approach. These results challenge the assumption that quality scales primarily with model size and highlight the potential of co-designing models and inference algorithms for more efficient and capable text-to-image generation.

## REPRODUCIBILITY STATEMENT

To facilitate reproducibility and encourage further research in this area, we commit to making our implementation publicly available upon publication. This includes all search algorithms, verifier integration code, and experimental configurations used in our study. Our work relies entirely on publicly available models and datasets, with Infinity-2B and all verifier models (ImageReward, CLIP-Score, Aesthetic Score, and LLaVA-OneVision) being openly accessible to the research community.

## AUTHOR STATEMENT ON THE USE OF LARGE LANGUAGE MODELS

Large language models were utilized in this work solely for editorial refinement and grammatical corrections. All scientific content, including research conception, experimental design, data analysis, and interpretive conclusions, originates entirely from the authors without computational assistance.

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

# A  QUALITATIVE RESULTS: BASELINE VS. BEAM SEARCH

We present qualitative comparisons on T2I-CompBench++ between baseline generation and our beam search approach guided by LLaVA-OneVision. The search is conducted with a computational budget of 195 image evaluations (1,365 NFEs). Each row shows the text prompt, baseline generation, and the search-guided result that best satisfies the prompt according to the vision-language model evaluation.

| Text Prompt | Baseline | Search-Guided |
|---|---|---|
| Bee left of key |  |  |
| Bird next to refrigerator |  |  |
| Bird top of balloon |  |  |
| Fish near car |  |  |
| Five kites |  |  |
| Four lamps, four dogs |  |  |

Table 10: Qualitative comparison of baseline and search-guided generation (Part 1 of 5)

| Text Prompt | Baseline | Search-Guided |
|---|---|---|
| Four pens | | |
| Giraffe right of wallet | | |
| Green apple, red kiwi | | |
| Green frog, yellow fly | | |
| Green rose, blue tulip | | |
| Rubber tire, fabric pillow | | |

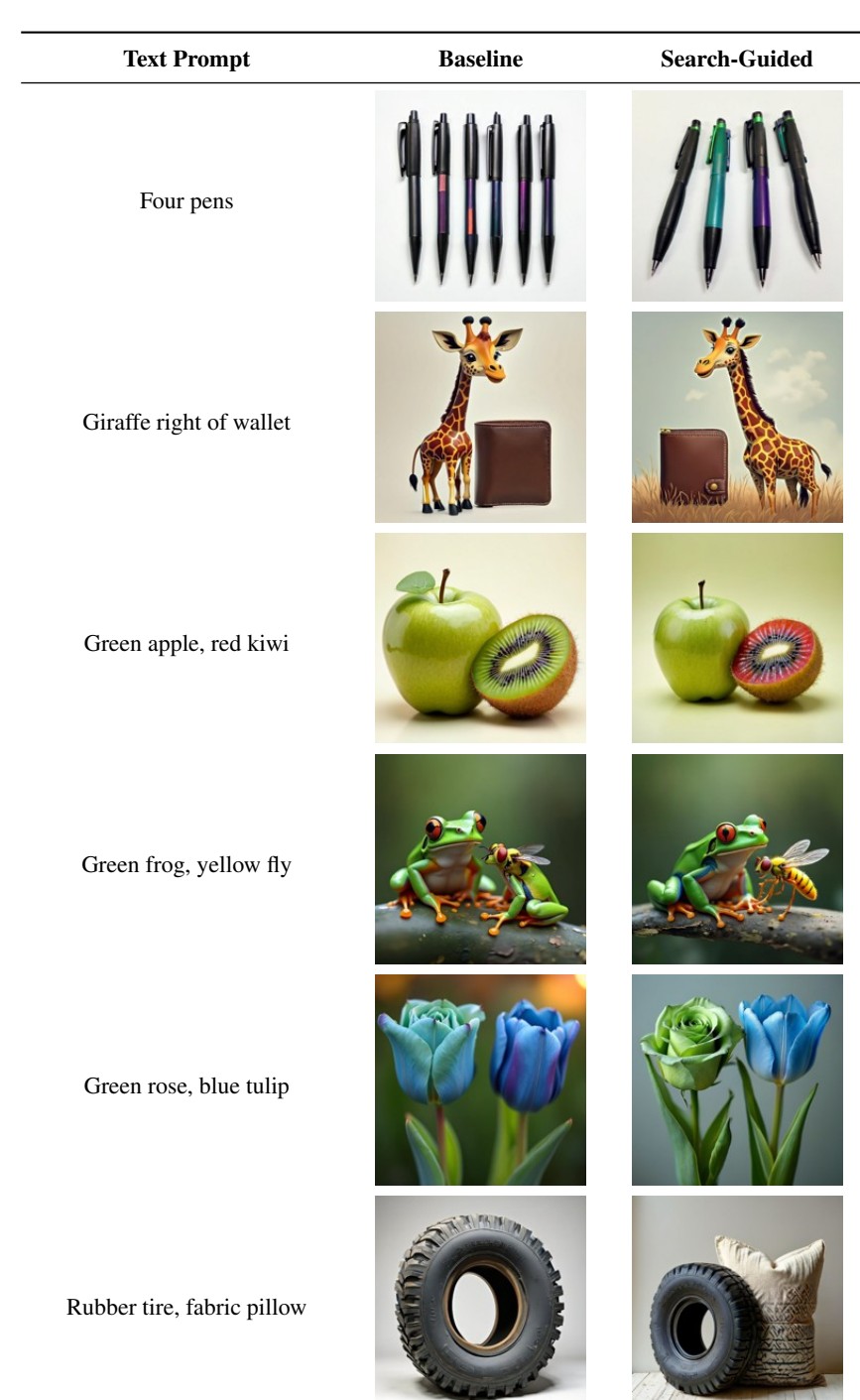

Table 11: Qualitative comparison of baseline and search-guided generation (Part 2 of 5)

| Text Prompt | Baseline | Search-Guided |
|---|---|---|
| Seven balloons | | |
| Six bread | | |
| Six ducks | | |
| Six keys | | |
| Small button, big zipper | | |
| Small lion, big horse | | |

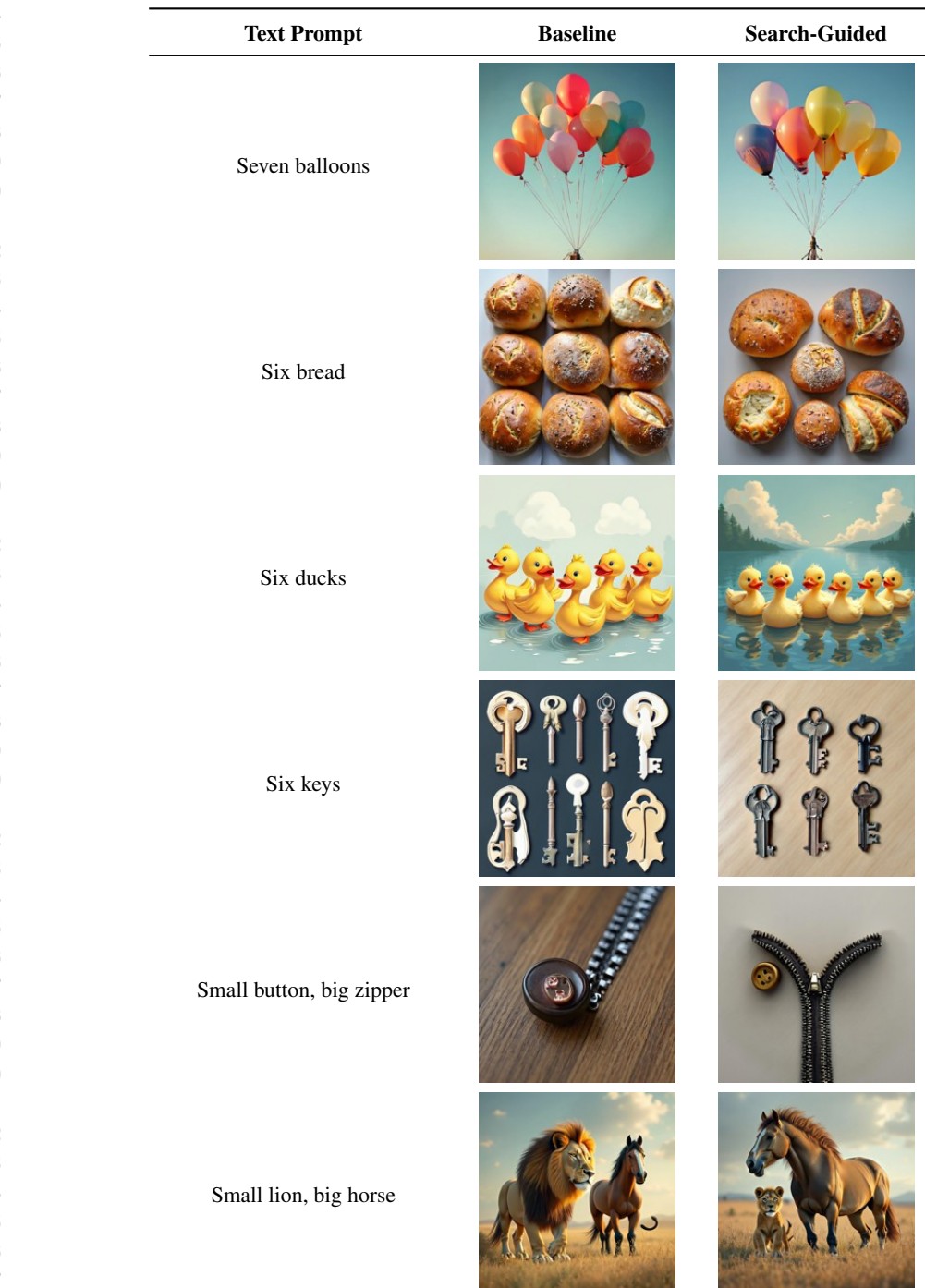

Table 12: Qualitative comparison of baseline and search-guided generation (Part 3 of 5)

| Text Prompt | Baseline | Search-Guided |
|:---:|:---:|:---:|
| Suitcase right of cow |  |  |
| Tall sunflower, short daisy |  |  |
| Three clocks |  |  |
| Two bowls, three microwaves, two chickens |  |  |
| Two sofas, four chickens |  |  |
| Two trains |  |  |

Table 13: Qualitative comparison of baseline and search-guided generation (Part 4 of 5)

| Text Prompt | Baseline | Search-Guided |
|---|---|---|
| Vase right of man |  |  |
| Woman right of TV |  |  |
| Wooden desk, leather jacket |  |  |
| Wooden fork, glass bowl |  |  |
| Wooden toy, fabric pants |  |  |

Table 14: Qualitative comparison of baseline and search-guided generation (Part 5 of 5)

## B RESULTS FROM DIFFERENT SEARCH STRATEGIES

| Prompt | Baseline (Single Sample) | Random search | Greedy Token Optimization | Beam search |
|---|---|---|---|---|
| *"A blue coloured pizza."* | | | | |
| *"A laptop on top of a teddy bear."* | | | | |
| *"A bird scaring a scarecrow"* | | | | |
| *"A zebra underneath a broccoli."* | | | | |
| *"A car playing soccer, digital art."* | | | | |

Figure 5: **Visual comparison of search strategies for text-to-image generation.** Each row shows results for a different prompt, with columns representing: baseline generation (single sample with Infinity-2B), random search, greedy token optimization, and beam search. All images are generated using the Infinity-2B model with identical parameters. All search strategies have been guided by the ImageReward (Xu et al., 2023) verifier. The budget is set to 390 verified images. All prompts are from Drawbench (Saharia et al., 2022).

# C SCALING BEHAVIOR FOR DIFFERENT VERIFIERS

Figure 6: **Performance scaling of search strategies on DrawBench across different verifiers.** The plots compare performance against computational budget (NFEs and images), showing that guided methods like GTO (blue) and beam search (green) are significantly more compute-efficient than random search (red). While both outperform random search at low budgets, beam search shows superior scaling, widening its lead over GTO as the budget increases.

| Selection Method | Imgs. | NFEs | Aesthetic | CLIPScore | ImageReward |
|---|---|---|---|---|---|
| Dynamic GTO | 130 | 1275 | 7.28 | 0.82 | 1.55 |
| Fixed GTO (high budget) | 195 | 1365 | **7.38** | **0.83** | **1.58** |
| Fixed GTO (low budget) | 130 | 910 | 7.24 | 0.82 | 1.54 |

Table 15: **Comparison of dynamic vs. fixed-budget GTO.** The variance-based dynamic allocation shows no clear benefit, proving less efficient than a fixed-budget approach for the same number of candidate images and underperforming a fixed-budget approach with a similar NFE count. Bold indicates best performance per metric.

## D   ANALYSIS OF A HEURISTIC-BASED DYNAMIC BUDGET ALLOCATION

A potential enhancement to tree search strategies is to allocate the computational budget dynamically, focusing resources on steps where the model is most uncertain. We hypothesize that the variance of verifier scores across candidate tokens would serve as a good proxy for this uncertainty. A high variance would suggest a complex decision point where more exploration is needed, while low variance would imply that the top candidates are similar and fewer samples are required.

**Experimental setup.** To test this hypothesis, we implemented a variance-based dynamic allocation strategy. The core of this method is a heuristic designed to distribute a fixed total number of candidate slots (130) across the 13 generation steps. The number of candidates allocated to each step was set to be proportional to the variance of verifier scores observed at that step, relative to the average variance across the entire generation process. As illustrated in Fig. 7, this approach methodically concentrates the search on the initial, high-variance steps.

**Results and analysis.** The performance of this strategy is compared against fixed-budget GTO in Tab. 15. The results reveal two key insights. First, when comparing methods that evaluate the same number of candidate paths (130 images), the dynamic approach is less compute-efficient. By front-loading the search, it consumed 1275 NFEs, 40% more than the 910 NFEs required by the fixed-budget strategy—without yielding a corresponding performance improvement (e.g., an ImageReward score of 1.55 vs. 1.54). Second, when compared to a fixed-budget GTO with a similar NFE count (1365 NFEs), the dynamic strategy still underperforms across all key metrics.

This result, while negative, provides a valuable insight: score variance alone appears to be a noisy or insufficient signal for optimal budget allocation. The upfront computational invest-

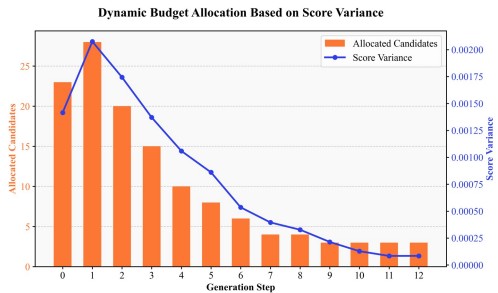

Figure 7: **Analysis of the variance-based allocation heuristic.** The plot shows the verifier score variance (blue line, right axis) at each generation step, which peaks early in the process. The orange bars (left axis) show the corresponding number of candidates allocated by the heuristic, mirroring the variance trend.

ment from the front-loaded dynamic strategy does not yield a proportional return in quality. One possible explanation is that even later, low-variance steps can contain critical decision points where a consistently wider search is beneficial. This finding suggests that a stable search breadth at each step (as in fixed-budget GTO) or the parallel exploration of multiple paths (as in beam search) are more robust and efficient strategies than this specific heuristic-based approach.

# E    VERIFIER IMAGE SELECTION

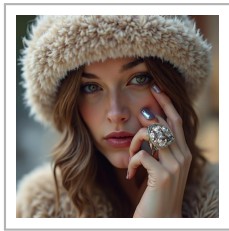 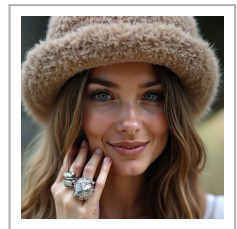 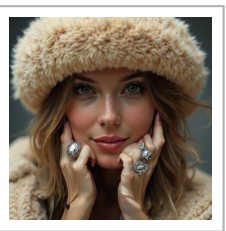

    (a) **Aesthetic**        (b) **CLIPScore**       (c) **ImageReward**      (d) **Ensemble**

Figure 8: **Search results for prompt "a metallic ring and a fluffy hat" using different verifiers.** The aesthetic verifier (a) produces visually appealing images but chooses an image without the metallic ring, demonstrating lack of prompt adherence. The other verifiers value prompt adherence, but does comprise aesthetic appeal by finding images that features poorly generated hands. Prompt and images from random search with 390 verified images using T2I-Compbench++ Huang et al. (2025)

