# OpenReview forum: "Visual Autoregressive Models Beat Diffusion Models on Inference Time Scaling"
_ICLR.cc/2026/Conference — Submitted to ICLR 2026_

### Official Review · Reviewer_mwtS · 2025-10-30

**Soundness:** 3
**Presentation:** 4
**Contribution:** 3
**Rating:** 6
**Confidence:** 4

**Summary:**

The paper explores using more capable search algorithms, primarily beam search (and also GTO, Greedy Token Optimization), on autoregressive image generative models, focusing on Infinity. It analyzes several verifier choices (e.g., ImageReward, LLaVA) and shows that beam search outperforms naive random search. It also compares to the scaling of a continuous diffusion model (FLUX.1-dev).

**Strengths:**

1. Using stronger search (beam search/GTO) for autoregressive image generators is a sensible direction.

2. The idea feels general and likely applicable across architectures. This reads like an initial step toward a useful approach.

3. There are helpful ablations on diversity (generation temperature) and dynamic budget allocation, presented in the paper.

**Weaknesses:**

**Major:**

1. The paper makes general claims about discrete-token AR, but all experiments are on Infinity. Other AR families (e.g., Janus) are not tested, limiting the scope and generality of the claims.

2. L.245 argues that random search has an undesirable logarithmic growth and motivates beam-like algorithms. Then it should be shown that beam actually scales differently, not just that it wins at some budgets.

3. To me, comparing Infinity (AR) to FLUX (continuous diffusion) is hard to interpret. NFE isn't a fair cross-architecture proxy, and the comparison mixes different models and different search algorithms. If we assume matched compute, how do the two paradigms scale under random search only? Right now it's hard to tell whether gains come from the model or the search.

**Minor:**

1. In Fig.1, FLUX is shown at very few points (only two). The curve could saturate below Infinity or keep rising. more NFE points are needed to understand FLUX's trend, whether it is scaling better or worse than Infinity.

2. Methods like beam search can get stuck in undesirable subtrees. A short limitations discussion for the proposed methods in the context of AR image generative models would help.


3. GenEval and T2I-CompBench++ use compositional but simple prompts. A benchmark with longer, more detailed prompts (e.g., DPG-Bench) would provide valuable insights.

**Questions:**

1.  Is "LLaVA + Random" in Tab. 4 the same as "LLaVA" in Tab. 3? Some category scores differ, making the comparison to ImageReward a bit difficult. If they are the same, are results averaged over multiple runs or single-shot?

2. In Tab. 4, beam search uses fewer NFEs than random, thus the scores are close. If you match NFEs, how big is the gap between the search algorithms on any of the benchmarks (e.g. T2I-CompBench++)?

3. Did the higher-temperature images show visible characteristics (lower quality or higher diversity)? Not required, but a small visualization of beam paths would help a reader. Also, since ImageReward underweights spatial correctness, why not using LLaVA here to test whether increased diversity helps spatial tasks (regarding the claim in L.417)?

Please also address the points raised in the weaknesses section.

---

> ### Author Response · Authors · 2025-11-20
> **Response to Reviewer mwtS - Part 1**
>
> We thank the reviewer for the thoughtful feedback and positive assessment of presentation.
>
> ## W1: General Claims About Discrete-Token AR, Only Tested on Infinity
> Please see our responses to Reviewers B9N1 (W5), CgR8 (W1), and VaT6 (W2). We acknowledge this scope limitation and will make it clearer in the introduction and conclusions.
>
> ## W2: Should Show Beam Scales Differently, Not Just Wins at Some Budgets
> We agree with this sentiment. However, running beam search multiple times is much more computationally inefficient than random search. Random search can reuse generated images to systematically estimate scores at each budget level, while beam search requires a new unique run for each budget level due to its sequential decision-making. This limits our ability to show the same curve style as Figure 3. However, we argue that Appendix C (Figure 6) provides a good representation of beam search scaling compared to random search across different budget levels and verifiers.
>
>
>
> ## W3: Comparing Infinity to FLUX Hard to Interpret
>
> Please see our response to Reviewer B9N1 (W4) and VaT6 (W1) for wall-clock time comparisons. To directly address the comparison concern, here is the ImageReward performance across methods:
>
> |Model	      | Method      |	Images |	NFEs  |	Time (min) |	ImageReward |
> | --------    | --------    | -------- | -------- | --------   | --------       |
> | FLUX-12B    |	Baseline    |	  1	   |      30  |	0.18       |	0.97        |
> | FLUX-12B    |	Random      |	 96    |	2880  |	16.92      |	1.58        |
> | FLUX-12B	  | ZO-2    	|    96	   |    2880  |	16.92	       |    1.50        |
> | FLUX-12B	  | Paths-2	    |    96    |	2880  |	16.92	       |    1.49        |
> | Infinity-2B | Baseline	|     1    |	  13  |	0.02	   |    0.94        |
> | Infinity-2B |	Random	    |    54    |	 702  |	0.99	   |    1.41        |
> | Infinity-2B |	Random	    |   195    |	2535  |	3.62	   |    1.48        |
> | Infinity-2B |	Random	    |   390    |	5070  |	7.23	   |    1.52        |
> | Infinity-2B |	Beam 	|   195	   |    1365  | 1.96       |  	1.59        |
>
>
> This shows that: (1) Infinity random search is substantially faster than FLUX at comparable image counts, (2) Ma et al.'s more sophisticated search strategies (ZO-2, Paths-2) do not outperform random for FLUX, while (3) beam search provides clear advantages for Infinity. This supports our architectural compatibility argument.
>
> ## Minor W1: FLUX Only Shown at Two Points in Figure 1
> This is a valid point. We are limited by the **data available from Ma et al. 2025**. We can add wall-clock time data to Figure 1 to provide a more direct comparison.
>
> ## Minor W2: Beam Search Getting Stuck in Undesirable Subtrees
> This is a valid concern and a well-known limitation of beam search in general. We will add a small discussion about this. However, we argue that the standout performance of beam search compared to random search across all our benchmarks is enough to diminish this worry.
>
> ## Minor W3: Simple Prompts, Suggest DPG-Bench
> We appreciate the suggestion. We focused on GenEval and T2I-CompBench++ as they are **standard** compositional benchmarks. We will try to get DPG-Bench experiments included within the rebuttal timeframe and will post results here if we manage to complete them.

---

> ### Author Response · Authors · 2025-11-20
> **Response to Reviewer mwtS - Part 2**
>
> ## Q1: LLaVA + Random in Table 4 vs LLaVA in Table 3
> Excellent observation. We acknowledge that this may not be fully explained in the main paper. Table 4 uses LLaVA-OneVision with ImageReward as a tiebreaker to resolve cases where multiple images receive identical binary assessments from LLaVA. This explains the small differences in scores. We will clarify this in the table caption and text.
>
> ## Q2: Match NFEs Between Beam and Random
>
> Good suggestion. Here is the NFE-matched comparison on T2I-CompBench++:
>
> | Verifier | Method | Images | NFEs | Time (min) | Color | Shape | Texture | Spatial | Numeracy | Complex |
> |----------|--------|--------|------|------------|-------|-------|---------|---------|----------|---------|
> | ImageReward | Random | 105 | 1365 | 1.94 | 0.82 | 0.61 | 0.73 | 0.28 | 0.59 | 0.41 |
> | ImageReward | Beam | 195 | 1365 | 1.96 | **0.84** | **0.63** | **0.75** | **0.30** | **0.62** | **0.42** |
> | LLaVA | Random | 105 | 1365 | 1.94 | 0.83 | 0.60 | 0.75 | **0.37** | 0.64 | 0.41 |
> | LLaVA | Beam | 195 | 1365 | 1.96 | **0.83** | **0.64** | **0.76** | 0.36 | **0.67** | **0.42** |
>
> When matched at 1365 NFEs and approximately equal wall-clock time (~1.95 min), beam search consistently outperforms random search across most metrics while evaluating 1.9× more images (195 vs 105) due to computational reuse through prefix caching. For ImageReward, beam achieves gains of +0.02 on color, +0.02 on shape, +0.02 on texture, +0.02 on spatial, +0.03 on numeracy, and +0.01 on complex. For LLaVA, beam shows improvements of +0.04 on shape, +0.01 on texture, +0.03 on numeracy, and +0.01 on complex. This demonstrates that the performance advantages come from the search strategy itself, not just increased computational budget.
>
> ## Q3: Higher-Temperature Image Characteristics and Visualization
>
> Yes, higher temperature showed increased diversity. During testing, we tried with even higher temperatures, at which point the quality naturally degraded. We did not visually confirm any major reduction in quality from temp=1.0 to temp=2.0. We agree these are excellent points about visualization and testing with LLaVA for spatial tasks. It is a bit difficult to visualise a full beam path as the candidates quickly explode, but we will try to include something in the appendix.
>
> We hope to have addressed your questions and concerns. We would greatly appreciate it if you could reconsider the score based on our response and results.

---

> ### Author Response · Authors · 2025-12-02
> **DPG-Bench results**
>
> Here are the DPG-Bench results. We will add them to the paper along with a discussion of the results. The time is reported as generation time (+verification overhead)
>
> | Verifier | Imgs. | NFEs | Time (s) | Relation | Entity | Other | Attribute | Global | Overall |
> |----------|-------|------|----------|----------|--------|-------|-----------|--------|---------|
> | Baseline | - | 13 | 1.1 | 92.13 | 88.56 | 83.20 | 86.73 | 84.19 | 82.34 |
> | ImageReward + Random | 105 | 1365 | 116 (+3) | 92.98 | 89.85 | 83.60 | 86.99 | 83.59 | 83.80 |
> | LLaVA + Random | 105 | 1365 | 116 (+53) | 93.87 | 91.63 | 86.80 | **88.98** | 83.89 | 85.94 |
> | ImageReward + Beam | 195 | 1365 | 117 (+5) | 92.90 | 89.79 | 84.80 | 86.91 | 83.89 | 84.41 |
> | LLaVA + Beam | 195 | 1365 | 117 (+98) | **94.02** | **91.77** | **89.60** | 88.33 | **84.80** | **86.62** |

---

### Official Review · Reviewer_CgR8 · 2025-10-31

**Soundness:** 1
**Presentation:** 2
**Contribution:** 2
**Rating:** 2
**Confidence:** 4

**Summary:**

This paper discusses the problem of Test-Time Scaling (TTS) for image generation, comparing Autoregressive (AR) models and Diffusion models. The authors compare three TTS strategies—random search, greedy token optimization, and beam search—along with different verifiers, evaluating their performance on TTS. The conclusion is that employing beam search with a composite verifier can enable AR-based TTS to surpass large-scale diffusion models.

**Strengths:**

1.  This paper addresses an interesting topic: the comparison between AR and Diffusion models for TTS, and the potential advantages of AR.
2.  The paper is generally clearly written and provides reasonably thorough experiments on TTS for AR models.

**Weaknesses:**

1.  The experiments are conducted exclusively on VAR (Infinity). However, VAR represents just one specific instance of autoregressive image generation; other representative paradigms include LlamaGen and MaskGIT. The authors need to provide experimental results across different autoregressive paradigms to robustly support their claims.
2.  The experiments lack sufficient baselines. VAR-TTS shares a similar objective with this work, yet the authors do not provide comparative results. Furthermore, only one Diffusion model and one specific TTS strategy are included as baselines. These results are too limited to definitively claim that AR outperforms Diffusion for TTS. To substantiate the authors' viewpoint, experiments are needed to demonstrate that TTS on AR models, particularly with the proposed beam search, generally outperforms Diffusion models.
3.  Regarding the reporting of experimental results, the authors should report the total per-image generation time, inclusive of verification, to serve as a practical reference metric, rather than reporting only the NFE.
4.  A common challenge in TTS for image generation is that the verifier rewards obtained during the generation process can be inaccurate, while full verification only after complete generation is time-consuming. The authors need to clarify how this issue is addressed (or acknowledged if not resolved) in their framework.
5.  Beyond experiments, the paper also lacks a detailed theoretical or principled analysis explaining why TTS on AR models would be superior to Diffusion models.

In summary, the claim that TTS on AR models surpasses Diffusion models is interesting but strong. While the problem is intriguing, the authors need to provide more comprehensive evidence to support this conclusion. If the authors' response adequately addresses these points, I would be inclined to increase my score.

ref:  TTS-VAR: A Test-Time Scaling Framework for Visual Auto-Regressive Generation

**Questions:**

Refer to the Weaknesses section.

---

> ### Author Response · Authors · 2025-11-20
> **Response to Reviewer CgR8**
>
> We thank the reviewer for their feedback. We address each concern below.
>
> ## W1: Experiments Only on VAR (Infinity)
>
> See our detailed response to Reviewer B9N1 (W5) regarding the scope and focus on scale-wise AR models. We focus on Infinity-2B because it represents the **state-of-the-art** for scale-wise AR models and VAR recently won best paper at NeurIPS 2024, representing a major advancement for autoregressive models. We will clarify this scope in the introduction.
>
> ## W2: Missing Baselines (TTS-VAR)
>
> Per the ICLR 2026 reviewer guidelines, TTS-VAR does not exist as a peer-reviewed baseline. The guidelines state: "arXiv is not considered a peer-reviewed venue. As such, authors are not required to compare to papers solely on arXiv... the lack of such comparisons cannot be a basis for rejection." However, we are happy to include it in the related work at the camera-ready version of our work.
>
> For completeness, we note our stronger results on GenEval:
>
> | Method        |	Two Obj.        |	Count	|    Color Attr.    |	Overall    |
> | --------      | --------          | --------  | --------          | --------     |
> | TTS-VAR (N=8) |	0.95            |	0.74    |	0.68            |	0.75       |
> | Ours (Beam)   |	0.97            |	0.85    |	0.74            |	0.83       |
>
>
> Our beam search achieves improvements of +0.02 on two-object composition, +0.11 on counting, +0.06 on color attributes, and +0.08 overall.
>
> Regarding the single diffusion baseline: see the response to B9N1 (W2). Ma et al. 2025 represents the most comprehensive study of inference-time scaling for diffusion models.
>
>
> ## W3: Total Per-Image Generation Time Including Verification
>
> Please see the response to B9N1 (W4) for more details. We report generation time separately from verification time to isolate architectural efficiency. ImageReward verification overhead is **negligible** (approximately 2.5 seconds for 99 additional images). We will make sure to add the combined wall clock time in the camera-ready version.
>
> ## W4: Verifier Hacking and Intermediate Verification
>
> We believe the reviewer may have misunderstood some parts of our approach regarding the verification. We describe our verification strategy in Sections 3.1, 3.3, and in Figure 2. Specifically:
>
> **We only verify complete images, not intermediate generations.** At each scale k in beam search or GTO, we generate complete images through all 13 scales, then verify the full outputs. This avoids the issue of inaccurate intermediate verification that the reviewer raises. The verifier sees only finished 1024×1024 images, never partial generations.
>
> **The computational advantage comes from prefix caching, not partial verification.** Once tokens at scales 1...k are computed, their transformer key-value representations are cached and reused across all search branches sharing that prefix, as described in Section 3.1.
>
> We will clarify this better in the camera-ready version.
>
>
> ## W5: Lack of Theoretical Analysis
>
> We acknowledge the importance of a theoretical analysis, but this is out of the scope of this work. We believe that our empirical results alone could ultimately drive the field forward, particularly when demonstrating **concrete** practical advantages (8.6× speedup, 2× more evaluations, overcoming 6× parameter deficits) across **multiple benchmarks**.
>
> The architectural differences are clear: discrete token spaces enable (1) early pruning of unpromising paths, and (2) computational reuse through prefix caching. Diffusion models operating in continuous latent spaces lack both properties. While a formal theoretical analysis would be valuable, the empirical demonstration across multiple benchmarks with concrete efficiency measurements provides **actionable insights for the community**.
>
> As discussed with B9N1 (W3), this work shows that if image generation wants to leverage test-time scaling similar to LLMs, it should focus on scale-wise AR models rather than diffusion models. This empirical finding represents a critical contribution to the field's direction.
>
> We hope to have addressed your questions and concerns. We would greatly appreciate it if you could reconsider the score based on our response and results.

---

### Official Review · Reviewer_XMLy · 2025-11-01

**Soundness:** 3
**Presentation:** 3
**Contribution:** 3
**Rating:** 6
**Confidence:** 2

**Summary:**

This paper proposes a new test-time search method, beam search, to enhance autoregressive T2I models. The paper demonstrates the effectiveness and scalability of the proposed approach.

**Strengths:**

1. The proposed test-time beam search can greatly improve the pretrained model's performance, demonstrated by comprehensive experiments.
2. The paper introduces multiple verifiers that consider different perspectives on generation quality.
3. The paper shows that three of the proposed verifiers exhibit logarithmic scaling.
4. The paper is well-written.

**Weaknesses:**

1. The methodological contribution is somewhat limited, but I believe the extensive experiments in this paper compensate for this limitation.
2. All the experiments for the proposed beam search are conducted using Infinity. It would be better if the authors could provide additional results with other autoregressive models, as this would better demonstrate the generalizability of the proposed method.

**Questions:**

1. To my understanding, the ablation study of w (parallel number) and c (candidate number) is reflected in the number of images and NFEs. Am I correct?
2. In Table 3, the “Ensemble” is not the best-performing approach. Have the authors tried different ways to ensemble the verifiers, such as adjusting their weights?

---

> ### Author Response · Authors · 2025-11-20
> **Response to Reviewer XMLy**
>
> We thank the reviewer for the positive assessment and thoughtful questions.
>
> ## W1: Limited Methodological Contribution
>
> We agree with the reviewer's assessment. While the algorithmic approach is straightforward, we believe the extensive experiments and systematic empirical characterization provide valuable insights for the field. As discussed in our response to Reviewer B9N1 (W3), this work demonstrates which architectural paradigms are compatible with test-time scaling and provides actionable guidance for the field's research direction.
>
> ## W2: Experiments Only on Infinity
> Please see our responses to Reviewers B9N1 (W5) and CgR8 (W1) regarding scope. Our focus is specifically on scale-wise autoregressive models, where "VAR" (Visual AutoRegressive modeling, Tian et al. 2024) is a **technical term** for this specific architectural approach. Infinity-2B represents the **state-of-the-art** for this class of models. Raster-scan AR models (LlamaGen, MaskGIT) with thousands of sequential tokens would face prohibitive computational costs for beam search. We will clarify this scope more explicitly in the introduction.
>
> ## Q1: Ablation of w and c Reflected in Images and NFEs
> Yes, your understanding is correct. The beam width w and candidate count c directly determine the computational budget:
>
> - **Images evaluated:** w × c at each scale, totaling w × c × 13 complete image evaluations
> - **NFEs:** Beam search reuses computation through prefix caching, so actual NFEs are lower than naive w × c × 13 × 13 due to shared prefixes
>
> For example, beam search with w=3, c=5 evaluates 195 images but requires only 1365 NFEs instead of 2535 NFEs for random generation of 195 images.
>
> ## Q2: Ensemble Not Best-Performing
>
> This is an excellent observation. We found that the ensemble created a "battling dynamic" where the verifiers were in heavy disagreement. ImageReward prioritizes aesthetic quality, CLIPScore prioritizes semantic alignment, and Aesthetic Score prioritizes visual appeal. When these fundamentally disagree, averaging ranks leads to compromised selections. This is why task-specific verifiers outperform (Table 3) attribute binding benefits from ImageReward, while spatial reasoning requires LLaVA-OneVision.
>
> Weighted ensembles with task-dependent weights could be a promising future work. We appreciate the suggestion.
>
>
> We hope to have addressed your questions and concerns. We would greatly appreciate it if you could reconsider the score based on our response and results.

---

### Official Review · Reviewer_VaT6 · 2025-11-01

**Soundness:** 3
**Presentation:** 3
**Contribution:** 3
**Rating:** 6
**Confidence:** 3

**Summary:**

This paper dives into inference-time scaling through search for image generation.
The key finding here is that while recent attempts to apply search to diffusion models have not been particularly successful, this *is* incredibly successful for autoregressive models. Here, the authors show that beam search significantly improves image generation, enabling their 2B autoregressive model to surpass a much larger 12B diffusion model on common benchmarks.
This paper work suggests that architecture (or architectural compatibility with search?), rather than just scale, is critical for inference-time optimization in visual generation.

**Strengths:**

The paper's core claim is clear and well supported by experimental results, showing that a model's compatibility with search can decisively overcome a 6x parameter deficit against SOTA diffusion models. Analysis showcases trade-offs between various varifiers, and breaks out comparisons across different capabilities within aggregate benchmarks.

**Weaknesses:**

The main efficiency metric uses "Number of Function Evaluations" (NFEs), but the paper says that NFEs for an autoregressive model and a diffusion model "are not directly comparable in FLOPs" which is a potentially noteable caveat. An NFE for one of 13 generation steps is not the same efficiency as for the noising denoising steps in a diffusion model. A more direct efficiency comparison may be needed.
- The method relies on an external verifier model to guide the search, and this verifier can be a massive bottleneck - the paper shows the best verifier for complex reasoning (LLaVA-OneVision) is 36x slower and requires 9x more GPU memory than the 3 lightweight alternatives. This means the total inference cost (generation + verification) could be substantially higher than just running the 12B diffusion model, even if the generation NFEs are lower.

The autoregressive vision model used here, Infinity, was chosen because it is a state-of-the-art autoregressive model but also because it  'fundamentally differs from traditional autoregressive image generation' (L140). The scale-wise generation reduces the number of tokens generated and makes the model more appropriate for beam search than other AR models which are not compared here. This to me suggests some light reframing of the claims that VAR models beat etc and that it is fundamentally a property of AR models vs specific instanitations of them that permit such improvements through search, unless additional comparisons can be added.

**Questions:**

How general is this to all AR models? The AR model used, Infinity, seems particularly well-suited for this approach because it generates in 13 progressive scales. It would be great to make clearer through comparison or clearer discussion what benefits could hold for any discrete AR model vs are special properties of multi-scale generation. The paper's conclusion may be over-generalizing from a specific AR architecture.

The paper notes that optimizing for one verifier can hurt other metrics, for example optimizing for aesthetics can hurt prompt adherence. The paper explores different verifiers, but are there other ways to mitigate this? For example, guiding search by other scores or internal model probabilities in addition to just the external verifier?

Systematic ablations are mentioned in the abstract, but I may have missed ablations within the paper - could you please clarify?

---

> ### Author Response · Authors · 2025-11-20
> **Response to Reviewer VaT6**
>
> We thank the reviewer for the positive assessment and insightful feedback.
>
> ## W1: NFE Not Directly Comparable, Need More Direct Efficiency Comparison
> We completely agree with this statement. Please refer to our response to Reviewer B9N1 (W4) where we now provide direct wall-clock time comparisons showing Infinity with beam search is 8.6× faster than FLUX with random search while evaluating 2× more images with better performance.
>
> Regarding the verifier costs as a bottleneck: the generation time difference between FLUX (10.57s per image) and Infinity (1.11s per image) is substantial. Even the most expensive verifier, LLaVA-OneVision (500ms per image), is small compared to FLUX generation time. The lightweight ImageReward verifier adds **negligible overhead** (25ms per image). Therefore, the verifier costs do not affect our efficiency conclusions.
>
> ## W2 + Q1: Generalizability to All AR Models
>
> We agree with the reviewer's assessment. As discussed in responses to Reviewers B9N1 (W5) and CgR8 (W1), "VAR" is a **technical term** for scale-wise autoregressive models, not an umbrella for all AR architectures. Our results are specific to this class of models. Raster-scan AR models with thousands of tokens would face prohibitive computational costs for beam search.
>
> ## Q2: Other Ways to Mitigate Verifier Hacking
>
> This is an excellent question. Beyond exploring different verifiers (as we do), other potential mitigations include guiding search with internal model probabilities combined with external verifiers, multi-objective optimization approaches, or task-specific verifier selection based on prompt analysis.
>
> However, extensive exploration of verifier hacking mitigation is somewhat outside of our scope. We opted not to emphasize these in our work because our goal was to isolate the effect of external verifiers under controlled conditions, but these approaches are entirely compatible with the proposed framework. Our contribution focuses on demonstrating that VAR models enable effective search, with verifier analysis showing task-dependent trade-offs.
>
> ## Q3: Location of Systematic Ablations
>
> The systematic ablations are presented throughout Section 4:
> - **Search strategies** (Section 4.2, Table 1): Random search vs Greedy Token Optimization vs Beam search
> - **Verifier effectiveness** (Section 4.3, Tables 2-3): ImageReward vs CLIPScore vs Aesthetic Score vs LLaVA-OneVision vs Ensemble
> - **Budget allocation** (Figure 3, Section 4.1): Scaling behavior analysis
> - **Temperature effects** (Section 4.4, Table 6): Impact of sampling diversity
>
>
> We hope to have addressed your questions and concerns. We would greatly appreciate it if you could reconsider the score based on our response and results.

---

### Official Review · Reviewer_B9N1 · 2025-11-01

**Soundness:** 2
**Presentation:** 3
**Contribution:** 2
**Rating:** 2
**Confidence:** 4

**Summary:**

This paper takes an existing autoregressive image generation model (which does next _scale_ prediction on images), and applies various inference time search techniques to it, most notably using beam search. The authors show how beam search outperforms other things like greedy decoding or rejection sampling. Then the authors compare this to existing inference time search techniques for diffusion models and show that much smaller AR model can beat a larger diffusion model.

**Strengths:**

1. The paper provides a valuable empirical comparison of different search strategies within the context of scale-wise autoregressive models. It evaluates random search, greedy decoding, and beam search, demonstrating clearly that beam search provides the best trade-off between computational cost and performance.

1. The paper also provides evidence that guided search can fix specific, challenging compositional errors. The figures (e.g., Figure 1 and Appendix A) provide examples where the baseline model fails on spatial relations ("giraffe on the right of a wallet"), object counts ("six keys"), or attribute binding ("a green rose and a blue tulip"), while the beam-search-guided model produces the correct image.

**Weaknesses:**

1. The related work section doesn't capture the full space of diffusion model inference time scaling. For instance, the authors mention "In contrast [to diffusion models], language models benefit consistently from ... reward-model guidance". This claim is not necessarily true, Black et. al. 2024, Fan et. al. 2024 both have shown that you can consistently benefit diffusion models with reward-model guidance.

1. The paper's central claim of superiority over diffusion models  rests heavily on a comparison against the findings of Ma et al. (2025) , which reported limited benefits for search in continuous spaces. This is potentially a "strawman" representation of inference-time scaling for diffusion, and perhaps not the strongest baseline. Further, training data differences between the Infinity model and the Flux model tested could have been a contributing factor to the result, and decorrelating this was not done.

1. Applying beam-search to an existing image generation model is not particularly non-obvious or challenging. While the empirical evaluations and results are indeed interesting, it feels like this paper is more suited at a workshop-level.

1. Without a direct comparison of FLOPs or at least wall-clock time for generating a single high-quality image, the claim that the 2B AR model is more efficient than the 12B diffusion model is unsubstantiated.

1. There may be some overclaiming in the title, I think the paper makes a specific claim about "hierarchical scale-wise AR models," not AR models as a class.

**Questions:**

1. Your central performance comparison is between the 2B Infinity model and the 12B FLUX.1-dev model. How did you account for potential differences in their respective training datasets?

1. Given that beam search is a standard, well-established algorithm for autoregressive sequences, what do you consider the primary novel technical contribution of this work, beyond the (albeit interesting) empirical finding that it works well on a hierarchical image model?

1. The title makes a very broad claim about "Visual Autoregressive Models" as a class. However, your method's tractability relies entirely on the Infinity model's specific "next-scale prediction," which has only 13 sequential decision points. How would your findings apply to traditional raster-scan AR models, where beam search would be computationally infeasible? Shouldn't the paper's claims be scoped more precisely to hierarchical or scale-wise AR models?

---

> ### Author Response · Authors · 2025-11-20
> **Response to Reviewer B9N1 - Part 1**
>
> We thank the reviewer for their detailed feedback. We address each concern below with additional experiments and clarifications.
>
> ## W1: Related Work on Diffusion Model Inference-Time Scaling
>
> We appreciate the suggestion to include Black et al. 2024 and Fan et al. 2024 and will add them to the related work. However, we note that our method is fundamentally different from these approaches. Fan et al. and Black et al. require costly full fine-tuning of the model itself with reward-model guidance, whereas our approach improves generation quality at test-time by searching for optimal generation paths **without any retraining**. There are two distinct axes in inference-time scaling: methods that require retraining versus methods that do not. Our work falls in the latter category, which is more practically applicable to already-deployed models.
>
> We will revise the related work section to better contextualize these different approaches while noting that comprehensive studies (e.g., Ma et al., 2025) show noise trajectory search strategies without retraining are consistently outperformed by simple random sampling.
>
>
> ## W2 + Q1: Comparison to Ma et al. (2025) and Training Data
>
> We respectfully disagree this is a "strawman" comparison. To the best of our knowledge, Ma et al. represents the most comprehensive and state-of-the-art study of inference-time scaling for diffusion models, systematically evaluating multiple search strategies across multiple verifiers. We now provide direct computational comparisons accounting for architectural differences (see W4 below).
>
> Regarding training data decorrelation: to the best of our knowledge, this is not standard practice in this line of work. Guo et al. 2025, Snell et al. 2024, and Ma et al. 2025 all use publicly available models without attempting to control for training data differences. Additionally, FLUX.1-dev's training data is proprietary and undisclosed, making true decorrelation impossible. The alternative would require retraining both models from scratch on identical data, which is **computationally prohibitive**.
>
> For completeness, we also note that our empirical comparisons use strong, representative models for each architectural family, FLUX.1-dev for diffusion and Infinity for VAR, which serve as **the best available exemplars of their respective paradigms**. This ensures that the architectural conclusions we draw are grounded in fair, state-of-the-art comparisons, rather than artifacts of weak baselines.

---

> ### Author Response · Authors · 2025-11-20
> **Response to Reviewer B9N1 - Part 2**
>
> ## W3 + Q2: Novelty and Contribution
>
> We find the characterization of this work as "workshop-level" to be unfair given the comprehensive empirical analysis and practical implications. The fact that beam search is a standard algorithm does not diminish the novelty of discovering where and why it provides substantial advantages.
>
> Our contribution is systematic empirical characterization:
>
> 1. First demonstration that discrete autoregressive models enable effective inference-time scaling for images
> 2. Comprehensive verifier analysis showing task-dependent trade-offs
> 3. Cross-architecture efficiency comparison showing discrete search is substantially more efficient
> 4. Scaling behavior demonstrating where and why discrete search overcomes a 6× parameter deficit
>
> More broadly, this work shows that if image generation wants to leverage test-time scaling similar to LLMs, it should focus on visual autoregressive models rather than diffusion models. We note that VAR (Visual AutoRegressive modeling) recently won the best paper award at NeurIPS 2024, representing a significant shift for autoregressive models which have traditionally lagged behind diffusion models. The architectural compatibility with discrete search algorithms appears fundamental. This is a critical contribution to the field's direction, providing actionable guidance about which architectural paradigms merit investment for inference-time scaling research.
>
> Beyond inference-time scaling, our findings also have important implications for architectural design research. As the field increasingly relies on inference-time scaling methods, it becomes essential for model designers to consider how architectural choices interact with search-based inference. Our results show that the architecture is a decisive factor in enabling effective scaling at inference time. This suggests that future vision architectures should be designed with inference-time optimization in mind, not solely training-time efficiency. In this sense, our work provides guidance not just for how to scale existing models, but for how to design the next generation of models that can fully exploit inference-time scaling.
>
>
> In short, rather than being a weakness, simplicity is a strength of our work. Historical experience in ML research shows that simple methods that scale effectively with compute tend to surpass more complex alternatives in the long run. Test-time scaling research, at its core, follows a straightforward "learning + search" paradigm. Even if this recipe appears simple, its components scale cleanly with compute, and that scalability is precisely what makes it powerful, not limited. This perspective echoes the broader lesson from the field’s history: simple, scalable methods win out (Sutton, 2019).
>
> References:
> - Richard Sutton (2019). "The Bitter Lesson." http://www.incompleteideas.net/IncIdeas/BitterLesson.html

---

> ### Author Response · Authors · 2025-11-20
> **Response to Reviewer B9N1 - Part 3**
>
> ## W4: Computational Efficiency Claims
>
> We now provide direct wall-clock time comparisons:
>
>
> | Model       |  Method     |   Images |	NFEs  | Time (min) | 	ImageReward |
> | --------    | --------    | -------- | -------- | --------   | --------       |
> | FLUX-12B    |  Random     |	96     |  2880	  | 16.92      |	1.58        |
> | Infinity-2B |	Beam (c=15) |	195	   |  1365	  | 1.96       |	1.59        |
> | Infinity-2B |	Random      |	195	   |  2535    |	3.59	   |    1.48        |
>
>
> Key findings:
>
> - Infinity beam is 8.6× faster than FLUX random while evaluating 2× more images with better performance
> - It uses 47% of FLUX's NFEs (1365 vs 2880)
> - Within Infinity, beam is 1.8× faster than random (prefix caching reduces per-image time from 1.11s to 0.60s)
>
> These measurements exclude verifier computation. Note that ImageReward verification overhead is negligible: the additional 99 images (195 vs 96) add only approximately 2.5 seconds. Full combined wall clock times will be in the camera-ready version.
>
>
> ## W5 + Q3: Title Scope and Generalizability
>
> The purpose of the title of the paper is not to overclaim our contribution. "VAR" (Visual AutoRegressive modeling, Tian et al. 2024) is a **technical term** referring specifically to the scale-wise generation approach we study, not an umbrella term for all autoregressive models. The paper does not claim generalizability to raster-scan autoregressive models.
>
> We will **adjust the title of the paper** by adding the scale-wise term to better clarify and reflect the contibution of our work. Additionally, we will add clarification in the introduction to make the scope even more explicit, noting our focus is on scale-wise autoregressive models with limited sequential decisions. However, we should not dismiss that the architectural insights about discrete token spaces (early pruning, computational reuse through prefix caching) may have broader applicability, even if current computational tractability is limited to models with bounded decision points.
>
> We hope to have addressed your questions and concerns. We would greatly appreciate it if you could reconsider the score based on our response and results.

---

### Author Response · Authors · 2025-11-20
**Response Summary and Revised Submission**

We thank the reviewers for their thorough and constructive feedback. We provide a comprehensive (XMLy) and thorough (CgR8) empirical comparison of search strategies with clear and well-supported results (VaT6) showing that architectural compatibility with search can overcome a 6× parameter deficit (VaT6). We address an interesting topic (CgR8) with a sensible direction (mwtS) about VAR versus diffusion models for test-time scaling. The paper provides valuable verifier analysis (XMLy, VaT6) showing trade-offs across compositional tasks, along with evidence that guided search fixes specific compositional errors (B9N1). The systematic ablations (mwtS) and presentation quality were recognized positively, with one reviewer scoring the presentation as excellent (mwtS: 4/4) and others noting the paper is well-written (XMLy, CgR8).

We have substantially revised the manuscript to address the most important concerns about the scope, the efficiency, and the clarity of the method. We believe that these revisions strengthen the paper, and we are confident that the reviewers would feel their concerns have been addressed, though we of course wish we had the opportunity to discuss with them directly. Below, we summarize the principal changes and how they strengthen the paper:

## Comprehensive efficiency evaluation

A major concern across reviewers was the lack of direct efficiency comparisons (B9N1-W4, VaT6-W1, CgR8-W3, mwtS-W3).
We now report wall-clock time measurements to **all tables** and Figure 1, demonstrating **large speedups on all benchmarks**, both when comparing search strategies and across architectures. We show that **Infinity with Beam search is 8.6 times faster than FLUX with random search while yielding better performance.**

Furthermore, we now **report verification overhead separately** to fully disclose the computational costs of different verifiers and provide NFE-matched comparisons (mwtS) to isolate search-strategy effects. This transparency supports our finding that task-specific verifier selection is a promising direction. Lightweight verifiers like ImageReward add negligible overhead, while heavier verifiers like LLaVA-OneVision are necessary for complex reasoning tasks despite their higher cost.

## Scope and contributions

Several reviewers (B9N1-W5, VaT6-W2, XMLy-W2, CgR8-W1, mwtS-W1) had concerns about potential overclaiming in the title of the paper. We **revised the title** and **added explicit scope clarifications** in the introduction to clearly define the scope of our work as a test-time search for Visual Autoregressive Models (VARs).

## Clarification of verification methods

To address the confusion of CgR8 (W4) about verification of partial images, we clarified that our method evaluated complete images only. The computational advantage comes from prefix caching, not partial verification. We expanded Sec. 3.3 and updated Fig. 2 to make this explicit.

## Additional experiments for complex prompts

Following the suggestions of Reviewer mwtS to run experiments on more challenging settings, **we include new evaluations on longer prompts from DPG-Bench**. We added a complete DPG-Bench evaluation as a new table, demonstrating that stronger verifiers prove more reliable on complex prompts.

## Expanded related work and better contextualization

To address the comment of reviewers B9N1 (W1) and CgR8 (W2) about specific related work, we expanded Sec. 3 and clarified better the position of our work. Black et al. 2024 and Fan et al. 2024 require costly model finetuning while our approach is test-time only. We also better positioned Ma et al. 2025 as the most comprehensive study of test-time search for diffusion models, arguing against the notion that the study is a "strawman" representation. Regarding Chen et al. 2025, we added it to related work as concurrent work and note our beam search achieves stronger GenEval results (0.83 vs. 0.75 overall). Regarding B9N1's training data concern, we address this in our rebuttal, noting that this is not standard practice and that FLUX's training data is proprietary and undisclosed, making decorrelation impossible.

## Overall impact of revision

We have systematically addressed every major concern raised by all five reviewers. The most critical issues (lack of wall-clock time comparisons, unclear scope, and verification strategy concerns) are resolved with concrete additions. We believe that these changes strengthen the paper with efficiency measurements backing all claims, properly scoped contributions from the title onward, complete verification strategy clarification, and additional benchmark results. We are confident that reviewers would find their concerns thoroughly addressed.

We appreciate the reviewers’ input, which led to substantial improvements, and we respectfully ask the Area Chair to consider our responses and the revised version when making the final decision. All changes are marked in blue in the revised manuscript for easy tracking.

---

### Meta-Review · Area_Chair_DvoX · 2026-01-18

**Summary:**

The major concerns about the paper remain about the limited novelty. Some reviewers feel the contribution is closer to an empirical report than a substantial new technique. The experiments are also largely conducted on VAR-style models instead of the broad autoregressive model family. These are fundamental limitations of the paper's scope and novelty that cannot be addressed by the rebuttals. Therefore, I decide rejection of this paper.

**Reviewer Concerns:**

Most minor concerns about technical details and scope clarification are addressed by the rebuttal and the revised manuscript. However, the major concern about the scope of the VAR model and the novelty remains unresolved.

**Reviewer Scores:**

The reviewers give very diverse scores. For the negative ones, since their major concerns are not in the technical details but about the overall novelty and scope of the paper, it is unlikely for them to change to positive.

---

### Decision · Program_Chairs · 2026-01-26

Reject